# Heart Rate Variability and Chronotype in Young Adult Men

**DOI:** 10.3390/healthcare10122465

**Published:** 2022-12-07

**Authors:** Joseph D. Vondrasek, Shaea A. Alkahtani, Abdulrahman A. Al-Hudaib, Syed Shahid Habib, Abeer A. Al-Masri, Gregory J. Grosicki, Andrew A. Flatt

**Affiliations:** 1Department of Health Sciences and Kinesiology, Biodynamics and Human Performance Center, 11935 Abercorn St. Savannah, Georgia Southern University, Savannah, GA 31419, USA; 2Department of Exercise Physiology, College of Sport Sciences and Physical Activity, King Saud University, Riyadh 11451, Saudi Arabia; 3Department of Physiology, College of Medicine, King Saud University, Riyadh 11451, Saudi Arabia

**Keywords:** cardio-autonomic, circadian rhythm, parasympathetic, cardiovascular

## Abstract

Whether morning heart rate variability (HRV) predicts the magnitude of its circadian variation in the absence of disease or is influenced by chronotype is unclear. We aimed to quantify associations between (1) morning HRV and its diurnal change, and (2) morning HRV and a Morningness–Eveningness Questionnaire (MEQ)-derived chronotype. Resting electrocardiograms were obtained in the morning and evening on separate days in a counterbalanced order to determine the mean RR interval, root mean square of successive differences (RMSSD), and standard deviation of normal-to-normal RR intervals (SDNN) in 23 healthy men (24.6 ± 3.4 yrs; body mass index: 25.3 ± 2.8 kg/m^2^). The MEQ was completed during the first laboratory visit. Morning RMSSD and SDNN were significantly higher (*Ps* < 0.05) than evening values. Morning RMSSD and SDNN were associated with their absolute (*Ps* < 0.0001), and relative diurnal changes (*Ps* < 0.01). No associations were observed between HRV parameters and the MEQ chronotypes (*Ps* > 0.09). Morning HRV was a stronger determinant of its evening change than chronotype. Greater diurnal variation in HRV was dependent on higher morning values. Strategies to improve basal HRV may therefore support healthier cardio-autonomic circadian profiles in healthy young men.

## 1. Introduction

A circadian rhythm is a biological pattern of approximately 24 h that is present in all organic life systems that have internal and entrainable oscillations which continue without external cues [1]. In humans, the circadian rhythm reflects the complex interplay between internal and external factors such as light–dark cycles, feeding and fasting, and exercise [2,3,4]. As part of the larger biological network that expresses daily rhythms, the cardiovascular system is also entrained to these patterns. For example, blood pressure (BP) exhibits a daily pattern [5], characterized by a morning rise that is maintained throughout the day, and a nadir during nighttime [6]. Deviations from these entrained rhythms is associated with increased risk for cardiovascular disease. Indeed, in a relatively recent large prospective cohort study, it was demonstrated that those with higher nighttime BP and BP that increased during sleep were at greater risk for coronary artery disease and heart failure [7].

Implicated in circadian disturbances in BP is autonomic imbalance, characterized by withdrawn vagal (i.e., parasympathetic) activity and sympathetic predominance [8]. Heart rate variability (HRV) describes the oscillations between consecutive heart beats and reflects autonomic heart rate regulation [9,10]. Higher HRV is associated with vitality and resiliency, and supports adaptation to new stimuli [11], whereas suppressed indices of vagal-mediated and global variability have been associated with various pathological conditions. For example, lower HRV is both cross-sectionally associated with and prospectively predictive of hypertension [12]. Similarly, HRV is inversely related to blood glucose in persons with and without diabetes [13]. Intriguingly, HRV appears to vary as a function of circadian rhythms [9]. Specifically, HRV is shown to be lowest in the afternoon and highest in the morning [14]. Meanwhile, blunted circadian variation in HRV has been observed in clinical populations. For instance, relative to patients with diabetes and angina, healthy controls demonstrated higher 24 h HRV, and greater diurnal variation [15]. Moreover, attenuated diurnal variation in HRV is associated with reduced BP dipping in hypertensive adults [8]. The capacity for HRV to decline throughout the day may depend on a higher morning value, as cardio-vagal responsiveness to stress is more pronounced and favorable in individuals with higher resting HRV [16]. However, the association between morning HRV and its diurnal change among healthy individuals is unclear. 

Chronotype can be feasibly and simply evaluated using the Morningness–Eveningness Questionnaire (MEQ), which is the most common method of determining chronotype [17]. The MEQ estimates peak alertness, and groups people as “morning types” (M-types; survey total: 59–86), “evening types” (E-types; survey total: 16–41), or “neither types” (N-types; survey total: 42–58) [17]. Chronotype is important because the differing circadian alignments are associated with disparate health outcomes. For example, E-types were found to have a greater prevalence of type 2 diabetes compared with M-types [18]. In addition, when comparing chronotype quartiles, each incremental increase in eveningness from definitive M-type to definitive E-type is associated with increased odds of having cardiovascular, nervous, renal, and other disease comorbidities [19]. Since meal timing, evening activities, and sleep–wake patterns tend to differ between chronotypes, diurnal HRV profiles may also differ. In a recent systematic review of the relationship between HRV and chronotype, there was some evidence of an association, but many studies involved specific conditions such as shiftwork, induced stress, exercise, or sleep deprivation [20]. Thus, the association between HRV and chronotype under free-living conditions in healthy adults requires further study. 

Diseases contribute to a diurnal cardio-autonomic profile in which little difference exists between HRV during morning and evening hours. Moreover, lower morning HRV has been associated with inferior cardiovascular and metabolic biomarkers in healthy young adults [21], and lifestyle factors such as sleep- [22] and meal timing [23] may influence morning HRV. However, whether morning HRV predicts the magnitude of its circadian variation in the absence of disease or is influenced by chronotype is unclear. Therefore, the purpose of this investigation was to quantify associations between (1) morning HRV and its diurnal change, and (2) morning HRV and a MEQ-derived chronotype. We hypothesized that individuals with higher morning HRV would exhibit greater diurnal variation and tend toward morningness chronotypes. 

## 2. Materials and Methods

### 2.1. Study Design

We cross-sectionally assessed the MEQ chronotype and morning and evening HRV in healthy young men. Each participant completed two visits to the laboratory on consecutive days in the morning and evening in a counterbalanced order. The relationship between morning HRV parameters and their absolute and relative changes from morning to evening (i.e., diurnal variation) were quantified, as were associations between the MEQ-derived chronotype and morning HRV.

### 2.2. Participants

We recruited 23 apparently healthy men who were free from cardiovascular, renal, and metabolic conditions, and were not using any prescription medications. Participants were instructed to maintain usual activities of daily living but were asked to abstain from alcohol and caffeine (e.g., tea, coffee, etc.) for 24 h before data collection through completion of their participation. This study was approved by the institutional review board (IRB) of King Saud University (IRB No. E-19-3965) and conformed to the guidelines provided by the Declaration of Helsinki.

### 2.3. Laboratory Visits

The morning laboratory visit took place at 0800 after an overnight fast. The evening visit took place at 1700 in a 4 h post-prandial state to minimize the influence of food or beverages on cardio-autonomic regulation [24,25]. Written and informed consent was obtained at the first laboratory visit prior to HRV data collection for all participants. The MEQ-derived chronotype was assessed following HRV assessment on the first visit. 

### 2.4. Chronotype—MEQ

Participants completed the MEQ to determine their chronotype. Participants were categorized based on their MEQ results as follows: participants with values ranging between 16–41 were considered evening-type (E); 42–58 were considered neither-type (N); and 59–86 were considered morning type (M) [17].

### 2.5. HRV

HRV metrics were obtained as previously described [26,27]. Three electrodes were placed in a modified lead II configuration with the ground electrode placed at the midpoint of the left clavicle. Cardiac rhythm was recorded with a computerized ECG data acquisition device with 16 analog input channels (PL3516 PowerLab 16/35, ADInstruments Pty Ltd., New South Wales, Australia). Recordings were performed in a quiet room with dim lighting while participants rested in a motionless supine position. Use of electrical devices (e.g., smartphones) was prohibited. Participants were asked to breathe naturally and avoid swallowing during the recording. A trained research assistant supervised the procedure and ensured the participant was awake and quietly resting. After a 5-minute stabilization period, RR intervals were recorded for 15 min while the researcher monitored signal quality. From the 15 min recording, a 5 min segment with minimal ectopic beats and artifacts was selected for HRV analysis. Filtering of RR intervals for artifacts and ectopic beats was completed using customized software (LabChart v. 8.1.13 Windows, ADInstruments Pty Ltd., New South Wales, Australia) as previously described [26]. Time-domain parameters included the mean RR interval (reflects resting heart rate), standard deviation of normal-to-normal RR intervals (SDNN, reflects global variability), and root mean square of successive differences in RR intervals (RMSSD, reflects cardio-vagal control) [9]. These parameters were selected due to growing use in the field via portable and wearable devices [21,28]. Moreover, time-domain parameters are less confounded by respiration rate than spectral indices [29].

### 2.6. Statistical Analysis

Normality assumptions were assessed with Shapiro–Wilk tests. Differences between morning and evening HRV values were compared using paired *t*-tests (e.g., for the mean RR interval and SDNN) while non-parametric comparisons (e.g., for RMSSD) were made with Wilcoxon Signed-Rank tests. Data are presented as mean ± standard deviation or median (inter-quartile range) for non-parametric data. Pearson correlations (r) were used to quantify bivariate associations between morning HRV parameters and their diurnal absolute and relative changes, as well as between morning HRV parameters and the MEQ chronotypes. Correlation coefficients were qualitatively interpreted as trivial (<0.1), small (<0.3), moderate (<0.5), large (<0.7), very large (<0.9), and near perfect (<1.0) [30]. Multiple linear regression analyses were subsequently performed to predict absolute and relative changes in HRV parameters based on corresponding morning values while including age, BMI, and the MEQ chronotype as covariates. Normality of residuals was assessed via Shapiro–Wilk tests; variance inflation factors were used to assess collinearity; and homoscedasticity was assessed via visual inspection of scatterplots displaying residuals and predicted values. Statistical analyses were performed using JMP (Version 16, Cary, NC, USA), and *p* values < 0.05 were considered statistically significant.

## 3. Results

The MEQ results showed that 8/23 participants were E-types, 9/23 were N-types, and 6/23 were M-types. Participant characteristics are presented in Table 1. 

Morning RMSSD was significantly higher than evening RMSSD (57.8 [32.6] vs. 47.1 [26.0 ms], *p* = 0.008), and morning SDNN was significantly higher than evening SDNN (56.6 ± 24.1 vs. 46.9 ± 13.1 ms, *p* = 0.028). A similar but non-significant morning-to-evening reduction was observed for the mean RR (938.8 ± 136.2 vs. 905.6 ± 127.6 ms, *p* = 0.085). 

Bivariate analyses demonstrated significant associations between morning RMSSD and the absolute change in RMSSD (r = −0.799, *p* < 0.0001) (Figure 1a), and between morning SDNN and the absolute change in SDNN (r = −0.841, *p* < 0.0001) (Figure 1b). Though moderate in magnitude, the association between the morning mean RR and the absolute change in the mean RR was not statistically significant (r = −0.404, *p* = 0.056) (Figure 1c). Similarly, significant associations were observed between morning RMSSD and the relative change in RMSSD (r = −0.581, *p* = 0.004) (Figure 1a), and morning SDNN and the relative change in SDNN (r = −0.726, *p* < 0.0001) (Figure 1b). Though moderate in magnitude, the association between the mean RR and the relative change in the mean RR was not statistically significant (r = −0.395, *p* = 0.062) (Figure 1c).

Multiple regression analyses controlling for age, BMI, and the MEQ chronotype demonstrated that morning RMSSD and SDNN were significantly predictive of the absolute and relative change in RMSSD and SDNN, respectively (Table 2). Model residuals followed a normal distribution (*p* > 0.05); independent variables did not demonstrate excessive collinearity (VIF < 1.3); and the assumption of homoscedasticity was confirmed.

## 4. Discussion

The purpose of this investigation was to quantify associations between (1) morning HRV and its diurnal change, and (2) morning HRV and a MEQ-derived chronotype. The main findings were: (1) morning RMSSD and SDNN were strongly related to their respective absolute and relative changes, independent of age, BMI, and chronotype; and (2) the MEQ chronotype was not significantly associated with any of the morning HRV parameters.

We observed significant reductions in RMSSD and SDNN, and non-significant reductions in the mean RR from morning to evening. These trends are in line with a review of 26 original studies showing that HRV exhibits a circadian pattern characterized by peak RMSSD, SDNN, and the mean RR during early morning hours and a nadir in the afternoon [14]. However, the timing of our measurements (i.e., 0800 and 1700) do not capture peak cardio-parasympathetic activity associated with sleeping. Moreover, post-waking HRV assessment at the laboratory may be somewhat reduced by hypothalamic–pituitary–adrenal axis activation pursuant of waking [31] and light exposure [32], which may explain why we observed only small reductions in RMSSD and SDNN. In agreement with this assertion, Bonnemeier and colleagues reported that RMSSD was highest during early morning hours (0400–0600), decreased after waking (0800–0900), and decreased further and remained suppressed throughout the afternoon and evening in healthy adults [33]; previous investigations have reported a similar daily HRV pattern [34,35]. Accumulated mental and physical stimuli from daily activities [36,37] along with a rising core body temperature [38] likely contribute to reduced HRV parameters as the day progresses. 

Morning RMSSD (Figure 1a) and SDNN (Figure 1b) were associated with their absolute and relative diurnal changes. Thus, a novel finding was that healthy individuals with higher morning HRV exhibited greater reductions in the evening, whereas individuals with lower HRV maintained low evening values. Shaffer and colleagues suggest that variability within a physiological system is critical for flexibility or adaptability, and epitomizes healthy function [39]. One way of evaluating this flexibility is through models of acute stress induction and recovery [40]. For example, in an investigation comparing the acute stress response of healthy men with low (<35.5 ms) and high (>35.5 ms) RMSSD, the high group exhibited greater absolute and relative decrements in RMSSD during the stressors, yet still had higher RMSSD than the low group after recovery from the stimuli [16]. Furthermore, only the high group showed cortisol recovery after 20 min, and TNF-alpha recovery at 60 min post stress while the low group maintained levels similar to immediately post stress [16]. Thus, higher morning HRV may be indicative of superior plasticity to accommodate stress. Attenuated diurnal variation in HRV, owing to low basal cardio-vagal function, has also been observed among individuals with disease [41]. In an investigation comparing patients with stable angina, type 1 diabetes, and healthy controls, healthy controls had higher SDNN (90 vs. 68.3 vs. 45.2 ms for healthy controls, type 1 diabetes, and stable angina, respectively) and RMSSD (38.9 vs. 24.7 vs. 9.6 ms) during the day and exhibited a larger night–day difference in SDNN (∆24.0 vs. ∆8.4 vs. ∆5.1 ms) and RMSSD (∆25.1 vs. ∆4.5 vs. ∆3.5 ms) [15]. Thus, it is noteworthy that between 6 and 7 of 23 participants in the current investigation showed no evening-related decline in RMSSD and SDNN, respectively, despite being young and apparently healthy. This may be relevant because lack of diurnal variation in HRV seems to be involved in the attenuation of nocturnal blood pressure dipping [8], which unfavorably affects long-term risk for cardiac events [42]. Moreover, reduced HRV parameters are associated with increased blood pressure and aortic stiffness in young adults [21], and are independent risk factors for cardiovascular diseases and cardiovascular and all-cause mortality [43].

We did not observe statistically significant relationships between the MEQ chronotype and morning HRV parameters (*Ps* > 0.09), though slopes of the associations were directionally as predicted. Some previous work has noted a relationship between HRV and chronotype while others have not. For example, in a comparison of 12 M-type and 12 E-type male collegiate soccer players before a training session, M-type athletes had significantly lower heart rates and higher LnRMSSD in the morning compared with E-types [44]. Interestingly, there was no between-group difference in HRV before the evening training session, which may suggest that different chronotypes exhibit dissimilar diurnal HRV patterns. Indeed, in a 16-day observational study in which wrist-based accelerometry was used to characterize chronotype, M-type individuals had higher RMSSD at multiple points in the day compared with E-types [45]. Conversely, in an investigation of 33 healthy adults (19 N-type and 14 E-type), there was no significant difference in SDNN or RMSSD between the two groups [46]. However, it is noteworthy that the two groups were more similar in chronotype (i.e., primarily N-type and E-types) due to a limited number of M-types [46]. Lack of sufficient participants with strong inclinations towards M-type patterns in the current investigation may explain why our associations between HRV and the MEQ chronotypes were marginal and non-significant. 

Findings of the present study should be interpreted in the context of a few limitations. First, morning and evening HRV measures are not as comprehensive as 24 h Holter monitoring to assess diurnal HRV patterns. Moreover, at both laboratory visits, activity, food, and caffeine intake were controlled for, but daily life events were not. In addition, other markers of physical and mental health may mediate relations between the MEQ chronotypes and HRV, as well as changes in HRV throughout the day, but were not characterized. Further, inclusion of females and participants from diverse age ranges, as well as more M-type individuals, would increase the generalizability of our findings.

In conclusion, the current findings suggest that diurnal variation in HRV, assessed by morning-to-evening change values, is attenuated among young men with lower morning HRV. Chronotype, as assessed by the MEQ, non-significantly explained only a small portion of the variance (<13%) in morning HRV, though an inadequate proportion of M-type individuals may account for this null finding. Thus, strategies known to improve nocturnal or post-waking HRV (e.g., exercise, dietary changes, stress management) may support a healthier circadian profile among young men with low cardio-vagal function, which may reduce their risk for chronic diseases.

## Figures and Tables

**Figure 1 healthcare-10-02465-f001:**
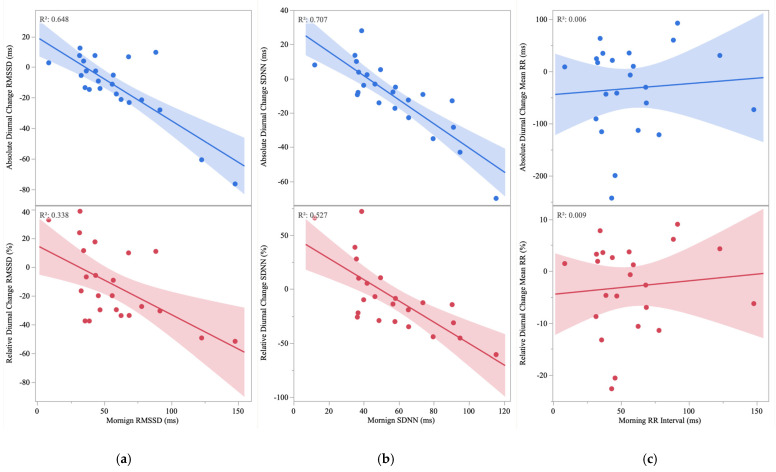
(**a**) Association between absolute and relative change in root mean square of successive differences in normal RR intervals (RMSSD) and morning RMSSD. (**b**) Associations between absolute and relative change in standard deviation of normal–to–normal RR intervals (SDNN) and morning SDNN. (**c**) Associations between absolute and relative change in mean R–R (mean RR) and morning mean RR.

**Table 1 healthcare-10-02465-t001:** Participant characteristics.

Participant Characteristics	Mean	±Standard Deviation
Age (years)	24.6	3.4
Height (cm)	172.9	6.9
Weight (kg)	73.4	10.1
BMI (kg/m^2^)	25.3	2.8
MEQ (au)	48.8	11.6

BMI = body mass index, and MEQ = Morningness–Eveningness Questionnaire.

**Table 2 healthcare-10-02465-t002:** Results of multiple linear regression models for absolute and relative changes in heart rate variability parameters.

	Model P	R^2^_Adj_	β_Std_	P
*∆RMSSD*	<0.001	0.590	-	-
MEQ			0.024	0.878
BMI			−0.126	0.389
Age			−0.028	0.840
Morning RMSSD			−0.811	<0.0001
*∆RMSSD%*	0.088	0.203	-	-
MEQ			0.087	0.692
BMI			−0.044	0.829
Age			−0.024	0.905
Morning RMSSD			−0.544	0.018
*∆SDNN*	<0.001	0.652	-	-
MEQ			−0.048	0.737
BMI			0.037	0.782
Age			−0.070	0.592
Morning SDNN			−0.823	<0.0001
*∆SDNN%*	0.004	0.466	-	-
MEQ			−0.176	0.322
BMI			0.145	0.386
Age			−0.013	0.931
Morning SDNN			−0.674	<0.001

RMSSD = root mean square of successive normal RR interval differences, SDNN = standard deviation of normal-to-normal RR intervals, MEQ = Morningness–Eveningness Questionnaire, and BMI = body mass index.

## Data Availability

Data is available upon request from the corresponding author.

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
