# Peer review of "Heart Rate Variability and Chronotype in Young Adult Men"

_healthcare, 2022, doi:10.3390/healthcare10122465_

Round 1
Reviewer 1 Report
I think it’s interesting research. I would recommend it if the following issues are handled properly.
1. Highlight the clinical significance.
2. What’s your main conclusion? I couldn’t find that easily.
3. Please explain the classification standard of chronotype that you are using in this research.
4. In Figures 1a and 1b, there are obvious correlations. Please explain that in detail.
5. Highlight the novelties in this research.
Reviewer 2 Report
Thanks for submitting this interesting work focused on the link between HRV patterns and circadian ones. I found the work timely and interesting, yet there are some key limitations that I would like the authors to address to unlock the contribution of the work.1) Some more information on inclusion/exclusion criteria are needed. eg drugs/coffee intake
2) It is unclear why you focused on time domain only for HRV and disregarded frequency domain features. This needs to be specified more.
3) Regardless of point 2 and the potential results please include to the very least information/data and add to your analyses on HF features as fundamental markers for the autonomic nervous system.
4) Key recent references are missing: Massaro, S., & Pecchia, L. (2019). Heart rate variability (HRV) analysis: A methodology for organizational neuroscience. Organizational research methods, 22(1), 354-393; Castaldo, R., Montesinos, L., Melillo, P., Massaro, S., & Pecchia, L. (2017). To What Extent Can We Shorten HRV Analysis in Wearable Sensing? A Case Study on Mental Stress Detection. In EMBEC & NBC 2017 (pp. 643-646). Springer, Singapore; Malpas, S. C., & Purdie, G. L. (1990). Circadian variation of heart rate variability. Cardiovascular research, 24(3), 210-213.
